# Optimizing immunization services: A Data Envelopment Analysis (DEA) of child immunization facilities in Pakistan

Taimoor Ahmad[1], Muhammad Ibrahim[2], Olan Naz[2], Mujahid Abdullah[1], Ayesha Khan[1], Maisam Ali[2], Elizabeth Bunde[3], Soumya Alva[3], Wendy Prosser[3], Adnan Ahmad Khan[2]*

**1** Akhter Hameed Khan Foundation (AHKF), Islamabad, Pakistan, **2** Research and Development Solutions (RADS), Islamabad, Pakistan, **3** International Division, JSI, Washington, DC, United States of America

* adnan@resdev.org

**Data Availability Statement:** All relevant data are within the manuscript and its Supporting Information files.

## Introduction

Child immunization, though cost-beneficial, experiences varying costs influenced by individual facility-level factors. A real-time solution is to optimize resources and enhance vaccination services through proper method to measure immunization facility efficiency using existing data. Additionally, examine the impact of COVID-19 on facility efficiency, with the primary goal of comprehensively assessing child immunization facility efficiency in Pakistan.

## Methods

Utilizing survey data collected in four rounds from May 2018 to December 2020, the research focuses on doses administered and stock records for the preceding six months in each phase. In the initial stage, Data Envelopment Analysis (DEA) is utilized to compute facility efficiency, employing two models with varied outputs while maintaining consistent inputs. Model 1 assesses doses administered, encompassing three outputs (pentavalent vaccine 1, 2, and 3). Meanwhile, Model 2, focuses on stock used featuring a single output (total doses used). The inputs considered in both models include stock availability, staff members, cold chain equipment, vaccine carriers, and vaccine sessions. The second stage involves the application of two competing regression specifications (Tobit and Simar-Wilson) to explore the impact of the COVID-19 pandemic and external factors on the efficiency of these facilities.

## Results

In 12 districts across Punjab and Sindh, we assess 466 facilities in Model 1 and 455 in Model 2. Model 1 shows 59% efficiency, and Model 2 shows 70%, indicating excess stock. Stock of vaccines need to be reduced by from 36% to 43%. In the stage, COVID-19 period reduced efficiency in Model 1 by 10%, however, insignificant in Model 2.

**Funding:** This work was supported, in whole or in part, by the Bill & Melinda Gates Foundation [grant number: INV-025171]. Under the grant conditions of the Foundation, a Creative Commons Attribution 4.0 Generic License has already been assigned to the Author Accepted Manuscript version that might arise from this submission. The funders had no role in study design, data collection and analysis, decision to publish, or preparation of the manuscript.

**Competing interests:** The authors declare that they have no competing interests.

**Abbreviations:** DEA, Data Envelopment Analysis; WHO, World Health Organization; IA2030, Immunization Agenda 2030; EPI, Expanded Program for Immunization; CCEOP, Cold Chain Equipment Optimization Platform; BHU, Basic Health Unit; RHC, Rural Health Center; THQ, Tehsil Headquarters; DHQ, District Headquarters.

## Conclusions

The proposed methodology, utilizing DEA, emerges as a valuable tool for immunization facilities seeking to improve resource utilization and overall efficiency. Model 1, focusing on doses administered indicates facilities low efficiency at average 59% and proves more pertinent for efficiency analysis as it directly correlates with the number of children vaccinated. The prevalent issue of overstocking across all facilities significantly impacts efficiency. This study underscores the critical importance of optimizing resources through the redistribution of excess stock with low efficiency.

## Introduction

The World Health Organization's (WHO) Immunization Agenda 2030 (IA2030) aims to reduce morbidity and mortality from vaccine-preventable diseases through increased vaccine coverage globally [1]. However, immunization coverage at scale is costly. For example, in 2018, Pakistan secured funding of USD 171 million for immunization, accounting for 7% of all public sector spending on health [2, 3]. Given that resources are always scarce, optimizing of available resources allows more children to be immunized with these limited resources, ultimately improving coverage. The need to optimize resources has been further highlighted in the aftermath of disruption of global supply chains and reduced demand for routine immunization during the COVID-19 pandemic [4].

The Expanded Program for Immunization (EPI) for child immunization was initiated in Pakistan in 1978 and immunizes children against childhood diseases such as pertussis, tuberculosis, poliomyelitis, tetanus, diphtheria, and measles [5, 6]. The current rate of 76% full immunization coverage at the national level is somewhat below the 81% global average (WHO 2021), and results in approximately 0.7 million children that are under or unimmunized [7]. Low uptake of immunization has been associated with inadequate service delivery resulting in poor service utilization [8], distances to EPI centers, inadequate outreach [9], inefficient utilization of funds [10], and unavailability of human resources or cold chain equipment [9]. The latter was addressed in a recent grant from the Gavi, the Vaccine Alliance.

A variety of techniques have been used to investigate efficiency in public health service delivery, such as WHO's National Immunization Program Reviews, to reveal components adversely impacting overall performance [11]. This and other techniques seek to explore waste or underutilization of resources, either by reducing the number of inputs to achieve minimal constant outputs–e.g., through a Data Envelopment Analysis (DEA) [12–18]–or to increase output levels while keeping the number of inputs fixed. DEA is employed to construct an empirical efficient surface for decision-making units (DMUs) with multiple inputs and outputs [19].

This methodology has found application in various sectors, including the banking industry and the education sector, and has been employed in consensus ranking [20–22]. While DEA has proven valuable in assessing the efficiency of healthcare facilities in recent years, its application to the evaluation of immunization facilities remains relatively limited [23–25], with no documented evidence from Pakistan. Additionally, the methodology has yet to be utilized in immunization facilities to analyze the optimal targets for maximizing input utilization.

This analysis presents the first application of DEA to evaluate the efficiency levels of child immunization facilities in Pakistan. We use data from EPI facilities, which were collected

during an assessment of the Cold Chain Equipment Optimization Platform (CCEOP) project, funded by Gavi, the Vaccine Alliance, which assessed the facilities in four different time periods where last assessment fell during the COVID-19 period. This created the additional possibility to conduct secondary analysis to explore the external impact of COVID-19 on the facilities' efficiency.

## Methodology

### Data collection

We use the data of EPI facilities from an assessment of the Cold Chain Equipment Optimization Platform (CCEOP) project, funded by Gavi, the Vaccine Alliance. The study team for this project included John Snow Incorporation (JSI) and Research and Development Solutions (RADS). Originally, the project was designed to estimate quality improvement through providing cold chain equipment to 136 EPI facilities and 4 district stores that had been randomly selected from a list of facilities and stores in the 12 districts of Punjab and Sindh provinces. Twelve facilities/district stores were selected at random from each district except Sujawal from Sindh, where we took the sample of eight facilities/district stores only. Inclusion criteria included those facilities/district stores who received the cold chain equipment during the project. Facilitates which were closed due to lack of vaccinations or any other reason were excluded from the sampling framework. The number of facilities/district stores to be interview were provided by JSI team given the cost consideration for this project.

The data collection process started from May 2018 and ended in December 2020 and included four cross-sectional surveys of same facilities. Data was collected through SurveyCTO which is computer assisted personal interview (CAPI) tool for tablet-based data collection. Verbal consent was taken from the facilities/district stores and recorded on SurveyCTO to document it before starting the interview. Survey data were collected in four rounds in May 2018, December 2018, December 2019 and December 2020 respectively. Data about doses administered and records of stocks were collected for 6 months prior to survey in each phase. In the first phase, recorded data was from October 2017 to March 2018 (T1). Second phase data was collected from May 2018 to October 2018 (T2). For third and fourth phases, data was recorded for the same months in two different years, June to November for 2019 (T3) and 2020 (T4) respectively. Since data collection for the last phase fell during the COVID-19 period, it also allowed assessment of the impact of the pandemic. As the survey did not involve any human subjects, hence, ethics approval was not required.

**Statistical analysis.** We use a two-stage approach to estimate the technical efficiency of immunization facilities and the impact of exogenous variables, particularly the COVID-19 period. The first stage includes estimation of technical efficiency using Data Envelopment Analysis (DEA) for immunization facilities using the necessary inputs and outputs. For immunization facilities, inputs are stock of doses available, number of cold chain equipment, number of staff members for immunization, number of days vaccines are administered (vaccination sessions) and number of cold boxes used for outreach (vaccine carriers), and outputs are the number of doses used (stock used) or number of doses administered (Table 1). We take these inputs because they are assumed to influence the vaccination process and have been used in prior studies [23, 26].

To identify which variables would be better suited as outputs in the first stage, we compare the doses administered and stock used variables. Logic dictates that total doses administered (Pentavalent vaccine 1 and 3) cannot be higher than stock used. However, we found such

**Table 1. Inputs and outputs for models 1 and 2.**

|  | Model 1 | Model 2 |
|---|---|---|
| **Inputs** | | |
| Stock available | × | × |
| Staff for immunization | × | × |
| Cold chain equipment | × | × |
| Vaccination carriers | × | × |
| Vaccination sessions | × | × |
| **Outputs** | | |
| Pentavalent vaccine 1 doses administered | × | |
| Pentavalent vaccine 2 doses administered | × | |
| Pentavalent vaccine 3 doses administered | × | |
| Stock used | | × |

discrepancies in 43 observations where doses administered numbers were higher than stock used. This stems from the fact that facilities occasionally acquire additional doses from other facilities, but which are not recorded in their ledger. For instance, if a facility is short of doses, then a vaccinator borrows doses from a nearby facility and records doses administered but does not record the corresponding additional stock in the stock ledger. Hence, the stock used numbers fall below doses administered. However, 22 facilities were dropped from the data because discrepancy was higher than expected (50 percent). While it might be possible that a facility may have borrowed more than 50 percent of its average stock of vaccines, there is even a higher chance of outliers or data issues that may have an adverse impact on the analysis; hence, dropping these facilities is preferred. The facility with missing observations for any input or output were dropped from the analysis by the DEA methodology.

The second stage captures the influence of exogenous variables, including catchment population, facility type, location and time period (COVID-19 last period), on efficiency scores using Tobit and Simar-Wilson regression specifications. The first and second stage models are explained in detail below. Stata 17 software is used for empirical analysis.

## Data Envelopment Analysis (DEA)

DEA is a non-parametric methodology using linear programming algorithm to compute the efficiency score for each DMU, i.e., immunization facility. Computation of each facility's technical efficiency is dependent upon other facilities, where most efficient facilities are using least number of inputs to maximize outputs and are set as a benchmark for others to describe the minimum number of inputs required for given set of outputs. In this way, the DEA calculates a measure of relative efficiency. For instance, if one facility is using more inputs for the same output level compared to the most efficient facility, its technical efficiency will drop. The most efficient facilities are given a value of 1 (100%) whereas the least efficient facilities are near the value of 0 (or 0%); as the value drops from 1 (100%) and it's closer to 0 (0%), the technical efficiency of the facilities decreases.

An advantage for using DEA is that it accommodates multiple outputs with different denominators. Hence, it makes DEA suitable for immunization facilities/health units as they produce multiple outputs within a facility using different inputs. Secondly, DEA methodology calculates weights after standardizing the inputs and outputs for all decision making units (immunization facilities in our case), and these weights allow to manage miscellaneous production functions which are not restrictive to corporate/industrial sector [27].

**First stage model specification.** We estimate two different models using DEA to calculate technical efficiency scores. Model 1 uses the first, second and third doses of pentavalent vaccine, while model 2 uses the total stock of pentavalent vaccine as outputs estimates. Both models have five inputs (Table 1). The model specifications are as below:

| | |
|---|---|
| *Maximize* | $h_k = \dfrac{\sum_{r=1}^{s} u_{rk} y_{rk}}{\sum_{i=1}^{m} v_{ik} x_{ik}}$ |

In the above equation, $h_k$ is defined as the relative efficiency of facility k. The numerator explains the weighted sum of outputs and denominator depicts weighted sum of inputs for facility k. The value of $h_k$ is censored between 0 and 1 as efficiency cannot exceed 100% and cannot be non-positive. $u_{rk}$ and $v_{ik}$ are the weights for each input and output which explain their relative importance while calculating efficiency.

We use the linear programming form of the primal Charnes, Cooper and Rhodes (CCR) model, in which denominator is set equal to 1 and the numerator is maximized [28]. An input-oriented model is adopted as immunization facilities are provided with predefined set of vaccines doses. The number of doses is determined in accordance with the catchment area of the facility and other resources are aligned. The DEA was run with constant returns to scale, as all facilities are homogenous in nature. The equipment is similar and same level of staff is dedicated for immunization services across all facilities. The final equations are as below:

| | |
|---|---|
| $Z_k = max \sum_{r=1}^{s} u_r \, y_{rk}$ | |
| Subject to: | |
| $\sum_{r=1}^{s} u_r \, y_{rk} - \sum_{i=1}^{m} v_i \, x_{ik} \leq 0$ | $k = 1, 2,\ldots, N$ |
| $\sum_{i=1}^{m} v_i \, x_{ik} = 1$ | |
| $u_r, \, v_i \geq 0$ | $i = 1, 2,\ldots, m \; r = 1,2,\ldots,s$ |

Where:

- $h_k$ *is the relative efficiency of facility k*

- $Z_k$ *is optimized value of efficiency indicator*

- $v_i$ *is the weights for input i*

- $u_r$ *is the weight for output r*

- $x_{ik}$ *is the input i used by facility k*

- $y_{rk}$ *is output r produced by facility k*

- *m is the total number of inputs*

- *s is the total number of outputs*

- *N is total number of facilities*

**Second stage model specification.** In the second stage, we estimate the effects of external variables on efficiency using Tobit and Simar-Wilson regression specifications [29]. Both these methods are based on censored data, which means dependent variable is fixed within a specific range. In case of efficiency estimates, the variable ranges from 0 to 1, making both regression models relevant.

Even though Tobit regression is used for censored data, it has some limitations in terms of efficiency variables. First, since the technical efficiency is calculated using DEA, the Tobit model lacks a clear theory of the data generating process. Furthermore, the Tobit regression assumes that efficiency scores are independent observations in the sample but in reality, these scores are calculated from a common sample of data. To adjust for these issues, Simar-Wilson proposes an alternate procedure by creating a data generating process which assumes truncated rather than censored regression technique. Moreover, applying the bootstrap method in the Simar-Wilson specification solves the issue of efficiency scores being correlated and produces unbiased standard errors and confidence intervals. Therefore, both methodologies are incorporated to analyze the variation in the results produced by each regression.

The empirical model specification is given as:

$$Y_{it} = \beta_0 + \beta_1 LOGX_{1it} + \beta_2 X_{2it} + \beta_3 X_{3it} + \beta_4 X_{4t} + \epsilon_{it}$$

Where:

- $Y_{it}$ is the efficiency score of facility i for the time period t

- $LOGX_{1it}$ represents logarithm of under two-year-old population of the facility catchment area

- $X_{2it}$ is a dummy variable for provinces (Punjab = 1 and Sindh = 2)

- $X_{3it}$ is a categorical variable for the type of facility (Basic Health Unit (BHU) = 1, Rural Health Center (RHC) = 2, Dispensary/Clinic = 3 and Hospital/Tehsil Headquarter (THQ)/ District Headquarter (DHQ) = 4)

- $X_{4t}$ is a categorical variable for the four-time periods (T1 = 1, T2 = 2, T3 = 3 and T4 = 4)

- $\epsilon_{it}$ is the residual term capturing the variance not included in the model

## Data transformation

For targeted population, we compute the proportion of children under two years of age out of the total population for each district from the population census of Pakistan [30]. After getting the proportion of children under-two for each district, we multiply it with the total catchment population (all-age groups) reported in the data. Thus, we impute the number of children under-two among the catchment population. We then take log-transformed values of catchment population to normalize it and reduce its skewness.

Our methodology focuses on outputs of facilities. Some data issues were analyzed while comparing the doses administered and stock used at facilities. Doses administered are the number of vaccines injected (separate numbers for each pentavalent vaccine 1 and pentavalent vaccine 3 administered, and not necessarily unique children vaccinated). In order to complete the doses administered data, pentavalent vaccine 2 is imputed taking the average of pentavalent vaccine 1 and 3. From national survey, it is clear that percentage of children receiving the first dose is high and percentage for the third dose is low, while the number of children vaccinated for second dose falls in between [31]. Whereas stock used is the aggregate of all doses administered, wastage, and stock given to neighboring facilities.

## Results

Table 2 presents summary statistics for all surveyed facilities across four distinct time periods. During the second period, there is a notable decrease in the average administration of Pentavalent doses. The Covid-19 period follows as the second lowest in terms of doses administered among the four periods. While stock availability is also at its lowest during the second period, the averages of all other inputs remain relatively consistent across the surveyed facilities.

Our DEA model shows that hospitals are the most efficient facility type for both doses administered (model 1) and stock used (model 2), whereas BHUs (0.57) are the least efficient facility type for doses administered (model 1) and dispensaries/clinics (0.69) for stock used (model 2). In both models, the T1 period (0.65 and 0.74) was the most efficient, and T2 (0.55 and 0.66) was the least. Mean efficiency is 59% for model 1 and 70% for model 2 (Table 3).

Apart from calculating efficiency scores, DEA allows us to suggest inputs adjustment for each facility type so that facilities can move towards 100% efficiency (Fig 1). The adjustment is recommended based on the reference group, which includes already-efficient facilities. The inputs adjustment shows that stock available is the most important input which should be reduced; even though the percentage difference of stock available is less than other inputs, the absolute difference is higher. Since model 2 has higher efficiency, its difference percentages are lower for all inputs on average as compared to model 1. After stock available, cold chain equipment seems to be the second most important input which needs reduction.

Since hospitals are the most efficient facility type in our sample, the input reduction (percentage difference) is lower for them on average as compared to other types of facilities.

**Table 2. Summary statistics for variables.**

| Variables | Overall Mean (CI) [32] | T1 Mean (CI) {2} | T2 Mean (CI) {3} | T3 Mean (CI) {4} | T4/Covid-19 Mean (CI) {5} |
|---|---|---|---|---|---|
| **Outputs/Inputs** | | | | | |
| Pentavalent 1 dose administered | 86.1 (81.3,90.9) | 88.2 (77.7,98.7) | 80.0 (69.9,90.1) | 90.1 (80.8,99.4) | 86.1 (77.3,94.8) |
| Pentavalent 2 dose administered | 80.74 (76.4,85.1) | 81.46 (71.8,91.1) | 74.62 (65.9,83.4) | 85.2 (76.7,93.6) | 81.54 (73.7,89.4) |
| Pentavalent 3 dose administered | 74.66 (70.7,78.7) | 74.3 (65.1,83.5) | 68.2 (60.6,75.8) | 79.6 (71.7,87.4) | 76.5 (69.1,83.8) |
| Pentavalent vaccine stock used | 245.5 (232.5,258.5) | 257.4 (228.2,286.6) | 221.6 (197.1,246.0) | 262.2 (234.9,289.5) | 240.8 (217.7,264.0) |
| Pentavalent vaccine stock available | 360.0 (341.1,378.9) | 367.3 (327.4,407.3) | 329.7 (296.0,363.4) | 393.2 (349.1,437.2) | 350.0 (316.9,383.1) |
| Staff for Immunization | 2.7 (2.5, 2.9) | 3.39 (2.8,4.0) | 2.5 (2.29,2.8) | 2.4 (2.1,2.7) | 2.6 (2.4,2.8) |
| Cold Chain Equipment | 1.7 (1.6,1.7) | 1.2 (1.1,1.3) | 1.8 (1.7, 1.9) | 1.5 (1.4,1.6) | 2.0 (1.8, 2.2) |
| Vaccination Carriers | 2.9 (2.7,3.1) | 2.0 (1.69,2.3) | 3.2 (2.8, 3.6) | 3.3 (2.8,3.8) | 3.0 (2.8,3.3) |
| Vaccination Sessions | 5.4 (5.2,5.5) | 5.5 (5.2,5.7) | 5.3 (5.0,6.0) | 5.4 (5.1,5.7) | 5.4 (5.1,5.7) |
| **External Variables** | | | | | |
| Population Under Two Years of Age | 3,212 (2,830,3,594) | 3439 (2428,4449) | 3287 (2210,4364) | 2836 (2554,3117) | 3275 (2936,3615) |
| *Province* | | | | | |
| Punjab | 0.5 (0.5,0.6) | 0.5 (0.4, 0.6) | 0.5 (0.4,0.6) | 0.5 (0.4, 0.6) | 0.5 (0.4, 0.6) |
| Sindh | 0.5 (0.4,0.5) | 0.5 (0.4, 0.6) | 0.5 (0.4, 0.6) | 0.5 (0.4, 0.6) | 0.5 (0.4,0.6) |
| *Facilities* | | | | | |
| BHU | 0.6 (0.5,0.6) | 0.6 (0.5,0.7) | 0.6 (0.5,0.7) | 0.6 (0.50,0.7) | 0.6 (0.5,0.7) |
| RHC | 0.1 (0.1,0.1) | 0.1 (0.1, 0.2) | 0.1 (0.1, 0.2) | 0.1 (0.1, 0.2) | 0.1 (0.1, 0.2) |
| Dispensary/Clinics | 0.19 (0.16,0.22) | 0.19 (0.13,0.26) | 0.20 (0.13,0.26) | 0.19 (0.13, 0.26) | 0.2 (0.1, 0.3) |
| Hospital/THQ/DHQ | 0.1 (0.1, 0.1) | 0.1 (0.06,0.2) | 0.1 (0.1,0.2) | 0.1 (0.1,0.2) | 0.1 (0.1,0.2) |

Note: Column 1 shows mean of all time periods combined. Pentavalent vaccine 1 doses administered, Pentavalent vaccine 3 doses administered, Pentavalent vaccine stock used and Pentavalent vaccine stock available were calculated by taking six-month average for each facility. While Pentavalent vaccine 2 doses administered was imputed using the average of Pentavalent vaccine 1 and 3 doses administered.

**Table 3. Doses administered and stock used efficiency statistics by facility type and time period.**

| Facility Type | Doses Administered (Model 1) | | | Stock Used (Model 2) | | |
|---|---|---|---|---|---|---|
| | N | Mean | Confidence Interval | N | Mean | Confidence Interval |
| **All** | 466 | 0.59 | (0.57, 0.61) | 455 | 0.70 | (0.69, 0.72) |
| BHU | 276 | 0.57 | (0.55, 0.60) | 271 | 0.70 | (0.68, 0.72) |
| RHC | 52 | 0.59 | (0.52, 0.64) | 50 | 0.72 | (0.66, 0.77) |
| Dispensary/Clinics | 84 | 0.60 | (0.56, 0.65) | 82 | 0.69 | (0.64, 0.73) |
| Hospitals | 54 | 0.65 | (0.60, 0.70) | 52 | 0.75 | (0.70, 0.80) |
| **Time Period 1** | | | | | | |
| T1 | 95 | 0.65 | (0.61, 0.70) | 96 | 0.74 | (0.71, 0.78) |
| T2 | 120 | 0.55 | (0.51, 0.58) | 114 | 0.66 | (0.62, 0.69) |
| T3 | 124 | 0.60 | (0.56, 0.63) | 121 | 0.71 | (0.68, 0.74) |
| T4 | 127 | 0.58 | (0.54, 0.62) | 124 | 0.71 | (0.68, 0.74) |

The inputs adjustment can also be viewed with respect to the four time periods in our sample (Fig 2). In the T1 period, the staff should be reduced by three in model 1 and two in model 2. Results suggest that staff should be reduced more as we move towards the endline. Cold

**Fig 1. Doses administered and stock used slacks in percentage by facility type. Note:** Above graph shows results from doses administered and stock used (as outputs). The x-axis shows the average units of each input. Actual is the average input used by the facility, whereas projected is the desired level calculated through DEA methodology. Percentages shown in the plot region are the average reduction rates for each type of facility.

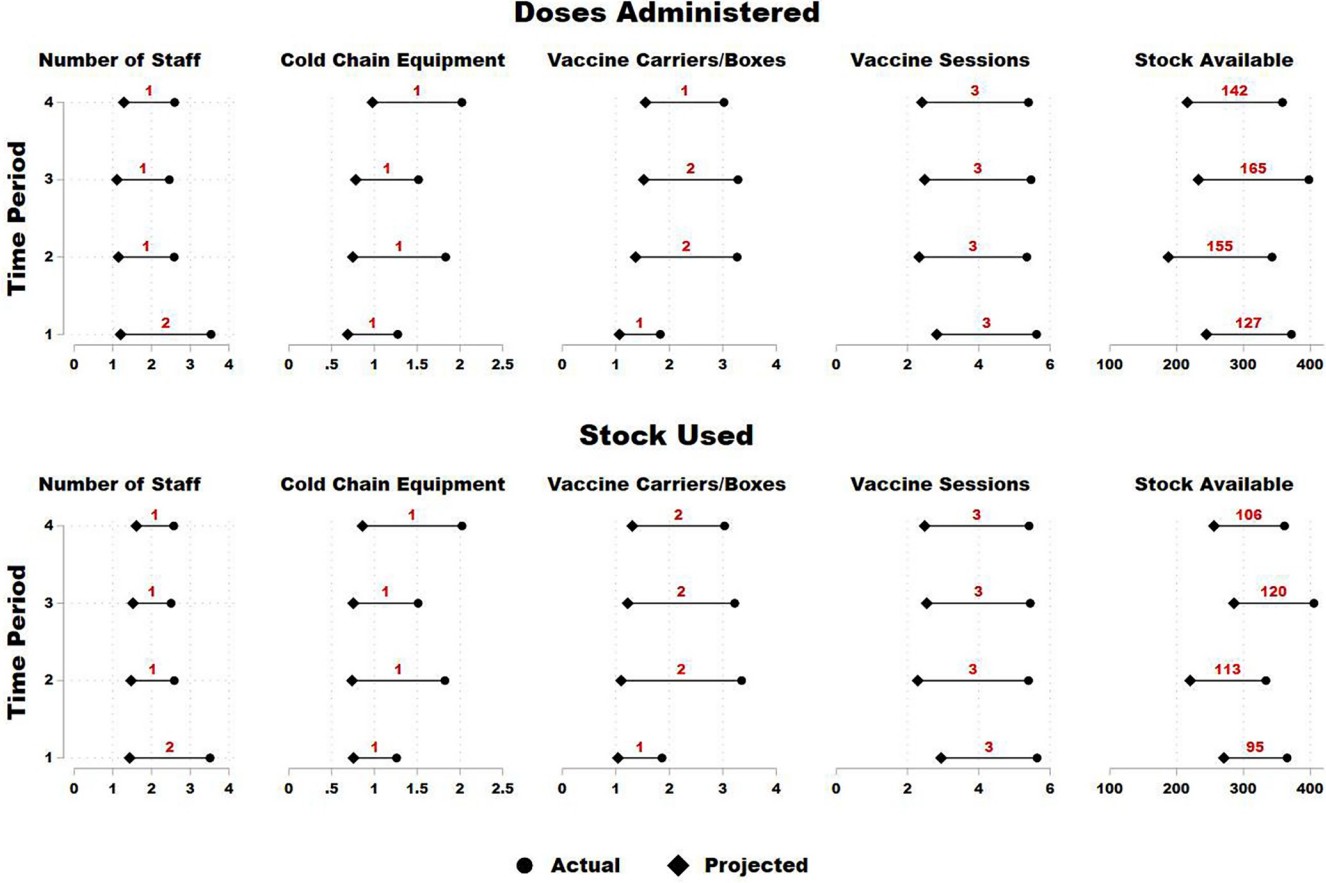

**Fig 2. Doses administered and stock used slacks in absolute values by time period. Note:** Above graph shows results from doses administered and stock used (as outputs). The x-axis shows the average units of each input. Actual is the average input used by the facility, whereas projected is the desired level calculated through DEA methodology. Numbers shown in the plot region are the average absolute number of units need to be reduced in each time period (1 denotes starting time period, likewise 4 is the last time period).

chain equipment should be reduced by one unit on average across all time periods. Vaccine carriers/boxes should be reduced by two units on average for both models across all time periods except in T1, where these need to be reduced by one unit. Vaccine sessions need to have a uniform decrease of three session per week for both models across all time periods. Lastly, stock available requires the highest absolute reduction out of all five inputs. The reduction in stock available is shown to be the lowest in T1 as that was the most efficient time period, while it is the highest for third time period (T3).

The second stage results (Table 4) show positive and significant relationship between efficiency and population (under two years old age group). As population grows around the facility, it is anticipated that the efficiency of the facility is likely to increase because higher population increases the percentage number of vaccinations administered. However, provinces show different coefficients for doses administered and stock used. Sindh is more efficient as compared to Punjab in model 1 in Tobit but insignificant in Simar-Wilson regression, while Punjab is more efficient in model 2 in both Tobit and Simar-Wilson regressions with 5% significance level. Facility type does not appear to be statistically significant across all regression specifications except for model 2 while using Simar-Wilson regression, where hospitals appear

**Table 4. Doses administered and stock used second stage regression results.**

| Efficiency | Doses Administered | | Stock Used | |
|---|---|---|---|---|
| | (1) | (2) | (3) | (4) |
| | Tobit Model | Simar-Wilson Model | Tobit Model | Simar-Wilson Model |
| Log under two years old population | 0.078* | 0.095* | 0.047* | 0.057* |
| | (0.047, 0.109) | (0.065, 0.124) | (0.020, 0.075) | (0.020, 0.094) |
| **Province [Base: Punjab]** | | | | |
| Sindh | 0.045* | 0.032 | -0.058* | -0.086* |
| | (0.005, 0.084) | (-0.002, 0.066) | (-0.093, -0.022) | (-0.132, -0.040) |
| **Facility Type [Base: BHU]** | | | | |
| RHC | -0.001 | 0.009 | 0.001 | -0.012 |
| | (-0.061, 0.060) | (-0.044, 0.062) | (-0.054, 0.055) | (-0.082, 0.059) |
| Dispensary/Clinic | 0.010 | 0.017 | 0.016 | -0.006 |
| | (-0.042, 0.062) | (-0.030, 0.063) | (-0.031, 0.063) | (-0.067, 0.055) |
| Hospital/THQ/DHQ | 0.052 | 0.039 | 0.053 | 0.089* |
| | (-0.011, 0.114) | (-0.018, 0.095) | (-0.002, 0.109) | (0.009, 0.169) |
| **Time [Base: T1]** | | | | |
| T2 | -0.125* | -0.088* | -0.084* | -0.082* |
| | (-0.180, -0.070) | (-0.136, -0.039) | (-0.133, -0.035) | (-0.147, -0.017) |
| T3 | -0.069* | -0.045 | -0.036 | -0.038 |
| | (-0.124, -0.015) | (-0.094, 0.004) | (-0.085, 0.012) | (-0.104, 0.028) |
| T4/COVID-19 period | -0.093* | -0.099* | -0.045 | -0.052 |
| | (-0.148, -0.039) | (-0.149, -0.049) | (-0.093, 0.003) | (-0.118, 0.015) |
| Constant | 0.031 | -0.137 | 0.398* | 0.364* |
| | (-0.215, 0.277) | (-0.369, 0.095) | (0.180, 0.615) | (0.077, 0.650) |
| Variance of efficiency | 0.040* | | 0.032* | |
| | (0.035, 0.046) | | (0.027, 0.036) | |
| Sigma | | 0.167* | | 0.195* |
| | | (0.155, 0.180) | | (0.175, 0.214) |
| Observations | 461 | 431 | 451 | 436 |
| Mean efficiency | 59% | | 70% | |

*Significant at 95% CI, 95% CI in parenthesis

to have a significant positive coefficient, suggesting that hospitals are more efficient as compared to BHUs.

The time variables show that efficiencies of facilities have dropped in all time periods as compared with T1 (baseline). However, the magnitude of the decline in efficiency varies across the time periods. Since efficiency is the highest in the T1 (baseline) time period, the succeeding time periods (T2 and T3) are associated with lower efficiency as compared to the baseline (see Table 3). The time variable also captures the T4/COVID-19 time period and it shows a negative relationship with efficiency for model 1 in both Tobit and Simar-Wilson regressions. It implies that efficiency is low in T4/COVID-19 period as compared to the T1 (baseline). Interestingly, T4/COVID-19 variable is not statistically significant for model 2 for both regressions.

## Discussion

Data Envelopment Analysis (DEA) of efficiency of immunization facilities in Punjab and Sindh shows efficiency to be 59% for doses administered (model 1) and 70% for stock used (model 2), suggesting that facilities have excess inputs by 44–47%. Furthermore, among the

facilities, hospitals are the most efficient which is expected because hospitals are mostly situated in densely populated areas. A second stage analysis indicates that COVID-19 reduced the efficiency of immunization facilities for doses administered (model 1) but the effect was insignificant for stock used (model 2), perhaps due to reduction in vaccination clients at the facilities. Given that most facilities have defined catchment populations that they serve, their outputs are relatively fixed. From a policy or program perspective, this analysis allows a means to understand when facilities are being oversupplied with inputs. Specifically, of the given inputs in our models, personnel and equipment are relatively fixed over time, however, how much vaccine is made available can be better adjusted using an efficiency analysis such as the one we describe.

Despite recent gains in vaccination coverage, Pakistan successfully vaccinates only around 76% of its children [31, 33] in part due to budget constraints, lack of major resources, and inefficient management [34]. In addition, it relies extensively on international donors for its Expanded Program for Immunization (EPI) [5]. An efficiency analysis can therefore serve as a simplified tool for facilities to promote optimal use of limited resources. In particular, it can help with optimal supply of vaccines to individual facilities where they can be used best. The best part of this methodology is that it may be automated through algorithms that can work with existing datasets such as the vaccine logistics management system and would not require inputs from statisticians. Furthermore, this analysis can be devolved from federal or provincial EPI to districts or even individual facility level.

Efficiency is an interplay between facility level outputs, i.e., the number of children vaccinated, and inputs, i.e., vaccine stocks, personnel, equipment and vaccination sessions [35]. Since most facilities have maximized the number of children they vaccinate in their catchment areas, including through outreach, efficiency may only be addressed by optimizing inputs [36]. Since personnel and equipment are often fixed for individual facilities or at least subject to only infrequent and long-term changes, stocks availability is the most flexible input that can be modulated. The analysis shows that DEA can be used to optimize the resources by identifying facilities overstocking and to maintain sufficient stocks, these facilities can reallocate to other facilities where stocks are needed. Conversely, our findings also suggest why interruption in supplies may not be immediately noticed by the personnel in the system as at least in the beginning, diminishing supplies may make the previously overstocked facility seem efficient. Additionally, given the issues of sporadic interruption in vaccine supplies, the program may want to identify the optimal level of efficiency that allows sufficient back up stores without wastage. DEA is the most applicable methodology in the literature to measure the technical efficiency of health facilities but immunization facilities are not covered extensively. However, given the scope of services provided by each health facility, inputs and output vary between them and across countries as well.

In the second stage, we have applied two regressions Tobit and Simar-Wilson to analyze the effect of external variables on efficiency of facilities. The magnitude of the coefficients is very similar and signs are consistent across both regressions. Since Simar-Wilson corrects for the bias resulting from correlation of efficiency values across facilities by bootstrapping them, it is suggested as the preferred regression method.

One of data collection periods (endline period in 2020) was affected by COVID-19 and therefore allowed us to measure its impact. In our analysis, COVID-19 has negative association with efficiency in model 1 (doses administered) while it is statistically insignificant for model 2 (stocks used). This is consistent with our above discussion that the number of children vaccinated is a primary driver of efficiency (in addition to stocks). Since fewer children could access facilities for vaccination during the period, efficiency fell [37, 38]. On the other hand, other observations suggest that transport of essential drugs and supplies such as vaccines was restored within a month in April 2020 (Khan et al., manuscript in process).

We have used two different model specifications that differ in terms of outputs only. Given this, we conclude that the preferred model is doses administered (model 1) as it does not over-state the output and represents accurate efficiency even though it is low. Stock used (model 2) overestimates efficiency due to increased output, which is another reason for insignificant effect of COVID-19. Overall, the immunization facilities need to improve their operational management as they are currently operating under par. A developing country like Pakistan needs to increase input utilization to generate maximum benefit from limited resources. The proposed methodology in this paper can serve as an instrument for resource allocation across all immunization facilities in Pakistan and effects of external variables can also be analyzed on the efficiency.

## Limitations

Despite the robustness of the study's results, several limitations have been identified, opening avenues for valuable future research. The initial constraint revolves around the limited dataset, specifically the absence of essential variables like energy consumption (electricity) that could potentially impact the immunization process. Additionally, the study relied on imputed data for pentavalent vaccine 2 doses administered, underscoring the need for enhanced recording and reporting practices. To address this, future research should consider comprehensive data collection to encompass a broader range of influential factors. Additionally, there is a need to validate the accuracy of catchment population figures, as the reliance on self-reported data from facility heads may introduce discrepancies. Enhancing accuracy through cross-referenc-ing with official records or alternative data collection methods is recommended. Beyond these data-related concerns, future research directions could explore additional factors influencing immunization facility efficiency, such as socio-economic variables, local infrastructure, and governance structures. By addressing these aspects, future research can significantly enhance the depth and applicability of findings, contributing to a more nuanced understanding of immunization facility performance.

## Conclusion

We use a two-stage DEA to measure efficiency of facilities using endogenous variables and then estimate the impact of exogenous variables. We employ two models which differ in terms of outputs which are doses administered and stock used. Facilities are operating at low effi-ciency levels of 59% and 70% for doses administered and stock used outputs respectively. This implies that facilities are allocated with excess inputs given their output level. Analysis high-lights the problem of overstocking in all facilities which can be utilized elsewhere. This meth-odology for the immunization facilities can play an important to optimize resources and improve overall efficiency level. In the second stage, we find negative impact of COVID-19 with efficiency majorly due to fear of parents to catch infection. In order to improve resource utilization and to increase the outreach of immunization facilities, sophisticated quantitative analysis should derive inputs allocation for each facility. Such methodologies could be rolled out to federal, provincial, district level or even facility level for routinely assessment of effi-ciency to get maximum benefit.

## Supporting information

**S1 Data.**
(DTA)

**S2 Data.**
(DTA)

**S3 Data.**
(DTA)

## Author Contributions

**Conceptualization:** Taimoor Ahmad, Muhammad Ibrahim, Adnan Ahmad Khan.

**Data curation:** Maisam Ali.

**Formal analysis:** Taimoor Ahmad, Muhammad Ibrahim, Olan Naz, Mujahid Abdullah, Ayesha Khan.

**Funding acquisition:** Adnan Ahmad Khan.

**Investigation:** Olan Naz, Ayesha Khan.

**Methodology:** Taimoor Ahmad, Muhammad Ibrahim, Olan Naz, Mujahid Abdullah.

**Project administration:** Adnan Ahmad Khan.

**Software:** Taimoor Ahmad, Muhammad Ibrahim.

**Supervision:** Ayesha Khan, Elizabeth Bunde, Soumya Alva, Wendy Prosser, Adnan Ahmad Khan.

**Validation:** Taimoor Ahmad, Muhammad Ibrahim, Olan Naz, Ayesha Khan, Elizabeth Bunde, Soumya Alva, Wendy Prosser.

**Writing – original draft:** Taimoor Ahmad, Muhammad Ibrahim, Mujahid Abdullah, Maisam Ali, Adnan Ahmad Khan.

**Writing – review & editing:** Olan Naz, Ayesha Khan, Elizabeth Bunde, Soumya Alva, Wendy Prosser, Adnan Ahmad Khan.

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
