## [Decision Letter · Decision Letter 0]

28 Nov 2023

PONE-D-23-14381Efficiency of Immunization Services and Impact of COVID-19: A Data Envelopment Analysis (DEA) of Child Immunization Facilities in PakistanPLOS ONE

Dear Dr. Khan,

Thank you for submitting your manuscript to PLOS ONE. After careful consideration, we feel that it has merit but does not fully meet PLOS ONE’s publication criteria as it currently stands. Therefore, we invite you to submit a revised version of the manuscript that addresses the points raised during the review process.

Please go through the Reviewers comments carefully and address them adequately.

We look forward to receiving your revised manuscript.

Kind regards,

Furqan Kabir

Academic Editor

PLOS ONE

Journal Requirements:

3. Please upload a new copy of Figures 1 and 2 as the detail is not clear. Please follow the link for more information: " ext-link-type="uri" xlink:type="simple">https://blogs.plos.org/plos/2019/06/looking-good-tips-for-creating-your-plos-figures-graphics/"
https://blogs.plos.org/plos/2019/06/looking-good-tips-for-creating-your-plos-figures-graphics

Reviewers' comments:

Reviewer's Responses to Questions

**Comments to the Author**

1. Is the manuscript technically sound, and do the data support the conclusions?

Reviewer #1: Partly

Reviewer #2: Yes

Reviewer #3: Yes

Reviewer #4: Partly

2. Has the statistical analysis been performed appropriately and rigorously? 

Reviewer #1: Yes

Reviewer #2: Yes

Reviewer #3: Yes

Reviewer #4: Yes

3. Have the authors made all data underlying the findings in their manuscript fully available?

Reviewer #1: Yes

Reviewer #2: Yes

Reviewer #3: No

Reviewer #4: No

4. Is the manuscript presented in an intelligible fashion and written in standard English?

Reviewer #1: Yes

Reviewer #2: Yes

Reviewer #3: Yes

Reviewer #4: Yes

5. Review Comments to the Author

Reviewer #1: The manuscript seeks to show a method for measuring the efficiency of an immunization program in a middle-income context using a data envelopment analysis method. The authors did a good job of justifying their analytic decisions. I have a few concerns and recommendations:

Major comments and suggestions

1. The abstract should be revised to ensure that it provides enough information for the reader to grasp the major issues discussed in the paper.

a. the abstract should provide clear and succinct information on the study context (location, study period), and the goal of the study. I

b. t should specify what kind of models are being described. Describing DEA before discussing models could work.

c. The result section of the abstract should include details like the overall numbers of health facilities included in the analysis.

d. The conclusion should focus on the policy or practical implications of the findings of the study.

2. The authors state that 22 facilities were dropped from the data. Facilities with missing observations were also dropped. It may be preferable to include these facilities in a sensitivity analysis to show what impact they have on the results. This is important given the lack of clarity on how facilities were randomized for inclusion, considering the acknowledgement that budgetary allocation influenced the extent of the survey.

3. The authors need to justify the choice of simple imputation for missing Penta 2 data. Why was multiple imputation which uses more variable not considered. The simple imputation resulted in a value for Penta 2 which was lower than both Penta 1 and 3.

4. It is unclear how the authors made the decision to combine data across the years for Table 2. It will be useful for the reader to see the disaggregated data for each year.

5. It does not seem to me that the authors accounted for usual vaccine wastage rate in the analysis. This a common feature of vaccination programmes and need to be factored onto efficiency calculations.

6. The suggestion that the findings of this analyses may be the basis for redeployment of resources need to be tempered given that a lot of potentially important factors are not considered in this analysis. For instance, demand-side considerations, equity and access issues all go into resource allocation decisions in public health programs. Furthermore, it is unclear from the description of the data collection period if year-round data was collected for the entire study period to account for potential seasonality in trends.

Other comments

1. The flow of thoughts on the paper can be improved. The services of a language editor may be helpful with this. For instance, paragraph 2 of the introduction mentions a “recent grant from the Gavi…” without an indication of whom the grant recipient is.

2. The sentence in the last paragraph of the introduction that reads, “…last assessment fell in the COVID-19 period” was unclear to me.

3. A lot of the material in the discussion session on model specification can be moved to the strength and weakness section of the discussion.

4. It is unclear to me what the numbers in brackets in the first paragraph of the results section mean.

Reviewer #2: This paper uses a two-stage DEA to measure efficiency of facilities using endogenous variables and then estimates the impact of exogenous variables. The topic is interesting. However, the paper needs carefully improvement in different sections.

1- Why do you use input-oriented DEA model is vague and needs more clarifions. Why not output-oriented one?

2- why do you use constant return to scale DEA model needs more explanations. why not variable return to scale one?

3- The validity of the proposed approach is vague. More analysis and discussion for the obtained results of the numerical example should be provided to prove the efficiency.

4- Author should update the introduction section deeply by stating the gaps of the existing study and then highlights the major results.

5- The literature review should be extended. Add and discuss papers with the application of the DEA in other fields. Please cite the following ones:

Equivalence relationship between the general combined-oriented CCR model and the weighted minimax MOLP formulation (2012) Journal of King Saud University-Science 24 (1), 47-54. An integrated data envelopment analysis and simulation method for group consensus ranking (2016) Mathematics and Computers in Simulation 119, 1-17. An interactive MOLP method for identifying target units in output-oriented DEA models: The NATO enlargement problem (2014) Measurement 52, 124-134. Fuzzy stochastic data envelopment analysis with application to NATO enlargement problem (2019) RAIRO - Operations Research 53 (2), 705721. A New Method for Solving Dual DEA Problems with Fuzzy Stochastic Data (2019) International Journal of Information Technology Decision Making 18(1), 147-170. Fuzzy Stochastic Data Envelopment Analysis with Undesirable Outputs and its Application to Banking Industry (2018) International Journal of Fuzzy Systems 20 (2), 534–548. A hybrid DEA-MOLP model for public school assessment and closure decision in the City of Philadelphia (2018) Socio-Economic Planning Sciences 61, 70-89.

Reviewer #3: Dear Author/Authors,

The manuscript has clear structure and purposes. However, I would like to make some suggestions to improve the study.

In the study, 4 different periods from 2018 to 2020 were discussed. For this reason, it was said that the effect of Covid was also examined. However, I see in table 3 that out of the 4 terms, only the second term has the lowest score. This already includes the period from May to October 2018. That's why the comments that it remained low due to the effect of Covid 19 are not very accurate. It should have been announced when there was a lower scoring period. I suggest changing the title. The study is not about Covid 19.

Additionally, it is wrong, in my opinion, to suggest that DEA's potential improvement recommendation is to reduce the number of employees. Because the number of employees is an uncontrollable variable. I recommend that an analysis be carried out taking this issue into consideration.

While defining the DEA model, the authors expressed both the objective function and the equality constraint as the entire DMU. However, this is not true. Only the DMU of interest should be included in the objective function and equality constraint. Therefore, the model is wrong and should be revised.

While you should use the same number of data for both models, for example, we see that there are 93 units in one and 96 units in the other for time period t1. These should have been the same. What is the reason why it is different? This causes misinterpretation when comparing average scores with each other.

Finally, the literature review should be expanded. Although the manuscript discussed many studies, it seems to be concentrated on a certain group of studies.

Reviewer #4: Inputs and outputs should be more and better clarified.

English language should be polished.

In this study, it has been stated "the most efficient facilities are given a value of 1 (100%) whereas the least efficient facilities are given the value of 0 (or 0%); as the value drops from 1 (100%) to 0 (0%), technical efficiency of the facilities decreases."

Has the efficiency score been obtained equal to zero in the considered DEA model?

In this research, it has been expressed "The value of ℎ is censored between 0 and 1 as efficiency cannot exceed 100% and

cannot be non-positive." Does mean it can take zero??

"u_r" ad "v_i" is correct. They should be investigated and revised.

Specify the units under review in detail.

Is the DEA model calculated for different years? Is not it better to use multi-period models?

Why the VRS has not been used instead of the CRS assumption.

Future research directions should be improved; in that, they should stem from the awareness of the limitations and opening avenues related to the obtained outcomes.

6. PLOS authors have the option to publish the peer review history of their article (what does this mean?). If published, this will include your full peer review and any attached files.

Reviewer #1: No

Reviewer #2: No

Reviewer #3: No

Reviewer #4: No

---

## [Author Response · Author response to Decision Letter 0]

12 Jan 2024

Thanks for the reviewing the manuscript and providing a detailed feedback. We have incorporated all the changes suggested which aligned with our work. Other questions have been answered according to our understanding and availability of data.

---

## [Editor Report · Decision Letter 1]

23 Jan 2024

Optimizing Immunization Services: A Data Envelopment Analysis (DEA) of Child Immunization Facilities in Pakistan

PONE-D-23-14381R1

Dear Dr. Khan,

We’re pleased to inform you that your manuscript has been judged scientifically suitable for publication and will be formally accepted for publication once it meets all outstanding technical requirements.

Kind regards,

Furqan Kabir

Academic Editor

PLOS ONE
---

## [Editor Report · Acceptance letter]

14 Mar 2024

PONE-D-23-14381R1 

PLOS ONE

Dear Dr. Khan, 

I'm pleased to inform you that your manuscript has been deemed suitable for publication in PLOS ONE. Congratulations! Your manuscript is now being handed over to our production team.

Kind regards, 

on behalf of

Dr. Furqan Kabir 

Academic Editor

PLOS ONE